# Downregulation of *LOC441461* Promotes Cell Growth and Motility in Human Gastric Cancer

**DOI:** 10.3390/cancers14051149

**Published:** 2022-02-23

**Authors:** Sang-soo Lee, JeongMan Park, Sooyeon Oh, KyuBum Kwack

**Affiliations:** 1Department of Biomedical Science, CHA University, Seongnam 13488, Korea; cconstant3383@gmail.com (S.-s.L.); jungman.park@chauniv.ac.kr (J.P.); 2Chaum Life Center, CHA University School of Medicine, Seoul 06062, Korea; ohsooyoun@hotmail.com

**Keywords:** gastric cancer, TNM stage, lncRNA, *LOC441461*, transcriptional modulation

## Abstract

**Simple Summary:**

Long noncoding RNAs (lncRNAs) are important regulators of cell progression or regression in gastric cancer (GC). The lncRNA *LOC441461* was downregulated in TNM stage IV GC. The downregulation of *LOC441461* promoted the proliferation, cell cycle progression, and metastasis of GC cells in vitro. We confirmed, with predictable targets, that *LOC441461* interacts with transcription factors and promotes tumor progression. Our study suggests *LOC441461* as a potential biomarker, which could also lead to a therapeutic target in patients with advanced TNM stage gastric cancer.

**Abstract:**

Gastric cancer is a common tumor, with a high mortality rate. The severity of gastric cancer is assessed by TNM staging. Long noncoding RNAs (lncRNAs) play a role in cancer treatment; investigating the clinical significance of novel biomarkers associated with TNM staging, such as lncRNAs, is important. In this study, we investigated the association between the expression of the lncRNA *LOC441461* and gastric cancer stage. *LOC441461* expression was lower in stage IV than in stages I, II, and III. The depletion of *LOC441461* promoted cell proliferation, cell cycle progression, apoptosis, cell motility, and invasiveness. *LOC441461* downregulation increased the epithelial-to-mesenchymal transition, as indicated by increased TRAIL signaling and decreased RUNX1 interactions. The interaction of the transcription factors RELA, IRF1, ESR1, AR, POU5F1, TRIM28, and GATA1 with *LOC441461* affected the degree of the malignancy of gastric cancer by modulating gene transcription. The present study identified *LOC441461* and seven transcription factors as potential biomarkers and therapeutic targets for the treatment of gastric cancer.

## 1. Introduction

Gastric cancer is a common malignancy, with a high mortality rate worldwide, and shows substantial incidence in East Asia, Eastern Europe, and South America [1]. The two most important risk factors for gastric cancer are *Helicobacter pylori* infection, which can induce atrophic gastritis, as well as intestinal metaplasia, and salted food intake [1]. *H. pylori* infection is the most prevalent, infecting more than 50% of all gastric cancers [2]. Common symptoms of gastric cancer include indigestion, anorexia, weight loss, and abdominal pain [1]. Gastric cancer with the persistence of symptoms before diagnosis can be incurable or in an advanced stage [1]. The diagnosis of gastric cancer is determined using the tumor, node, and metastasis (TNM) classification system, according to the American Joint Committee on Cancer [3]. Clinical staging is determined according to physical examination, biopsy, radiological imaging, and the results of an endoscopy [4,5,6]. In patients who undergo surgical resection, pathological staging can be performed, according to the analysis of surgical specimens [4,5,6]. The clinical and pathological staging of gastric cancer is determined by various combinations of the T, N, and M categories [7,8]. Although only 20% of patients survive more than 5 years after diagnosis, there is no universal standard first-line therapy [2,9]. For early gastric cancer with moderate differentiation or without invasion, upfront surgery is the preferred means of management. Perioperative or postoperative chemotherapy with chemoradiation is recommended in the current guidelines [10]. For the most advanced gastric cancer, a fluoropyrimidine chemotherapy and platinum-based doublet are typically the backbone regimen [9,10].

Long noncoding RNAs (lncRNAs) are longer than 200 nucleotides and transcribed from noncoding regions [11]. Although previous studies have focused on the role of mRNAs in gastric cancer, recent studies have highlighted the involvement of lncRNAs [12,13]. Due to this nature, lncRNAs have their own structural characteristics that play various roles in epigenetic regulation, transcriptional regulation, and post-transcriptional regulation [14]. LncRNAs regulate histone modifications at the chromatin level through histone methylation and acetylation, DNA methylation, and transcriptional activation, and function together with various transcription factors and post-transcriptional modifications [12]. The function of lncRNAs as a decoy molecule blocks the function of chromosome-folding proteins or transcription regulators [14]. In addition, lncRNAs associate with the regulation of gene expression through sponging miRNAs [14]. As guide molecules, lncRNAs lead specific molecules to target locations [14]. If the target molecules are transcription factors, lncRNAs have a role in cisregulation [14,15]. Alternatively, other lncRNAs play a role in transregulation with histone methylation [14,16]. LncRNAs also act as a scaffold that can recruit numerous molecules [14]. For example, a well-known scaffold lncRNA is an X-inactive specific transcript (Xist) that suppresses the expression of the X chromosome in females, and as a result promotes the assembly of polycomb complexes 1 and 2 [14,16]. From the perspective of mRNA processing, lncRNAs modulate splicing factors through regulating the phosphorylation of splicing factors, and hijacking them [16,17]. Based on the molecular function of lncRNAs, lncRNAs play different roles in development and disease, and are involved in the progression or regression of cancer, suggesting their potential as novel therapeutic targets [18,19]. For example, the expression of HOX transcript antisense RNA, which recruits chromatin modifiers, is associated with metastasis and a poor prognosis in lung cancer [20]. Metastasis-associated lung adenocarcinoma transcript 1 is a novel therapeutic target for decreasing resistance in prostate cancer [21].

In this study, we show that the lncRNA *LOC441461* is downregulated in clinical stage IV gastric cancer. The function of *LOC441461* has not been studied in detail, except in colorectal cancer, in which *LOC441461* expression is associated with cell growth and motility [22]. The role of *LOC441461* in modulating various cellular functions was investigated in human gastric cancer cell lines. The downregulation of *LOC441461* expression increased the aggressiveness of gastric cancer cells, suggesting that decreased *LOC441461* is associated with a poor prognosis.

## 2. Materials and Methods

### 2.1. Expression Data from TCGA

Clinical data, survival data, and RNA-seq data of STAD were downloaded from UCSC XENA (https://xena.ucsc.edu/ accessed on 10 December 2021). A total of 379 gastric cancer cases were analyzed for clinical TNM staging and RNA-seq data. The unit of RNA-seq data was log2(RPKM + 1).

### 2.2. Information Gain (IG)

Clinical TNM staging was performed for each individual. IG was assessed using the clinical staging of each individual, in addition to gene-level expression data using FSelector package v. 0.31 (a package in R, a language and environment for statistical computing, R Core Team, R Foundation for Statistical Computing, Vienna, Austria) [23,24]. Genes with an IG > 0 were used as the feature selection threshold for the clinical stage.

### 2.3. Survival Analysis

A survival analysis was conducted to investigate the difference in prognoses between groups, which were composed of a combination of the T and M categories. A logrank test was conducted with Survival v. 3.2 (https://cran.r-project.org/web/packages/survival/index.html accessed on 8 January 2022). The Kaplan–Meier plot was visualized with Survminer v 0.4.8 (https://cran.r-project.org/web/packages/survminer/index.html accessed on 8 January 2022).

### 2.4. Gastric Cancer Cell Lines and Culture

All human gastric cancer cell lines (AGS, SNU16, SNU216, SNU638, MKN45, and MKN74) were purchased from the Korean Cell Line Bank (KCLB; Seoul, Republic of Korea) and cultured in RPMI-1640 (Gibco, Grand Island, NY, USA) supplemented with 10% fetal bovine serum (FBS; Gibco) and 1% penicillin/streptomycin (Gibco). Cells were grown at 37 °C in a 5% CO_2_ incubator, under humidified conditions.

### 2.5. siRNA Transfection

The *LOC441461* siRNA oligonucleotide (si-*LOC441461*; BIONEER, Daejeon, Republic of Korea) and negative control scrambled siRNA (N.C, BIONEER) were transfected into MKN74 and SNU216 cells using Lipofectamine™ RNAiMAX (Invitrogen, Carlsbad, CA, USA) and Opti-MEM (Gibco) at a final concentration of 100 nM. After 24 or 48 h, cells were harvested for subsequent experiments. The oligonucleotide sequences of the siRNAs are listed in Appendix A.

### 2.6. Subcellular Fractionation and RNA Isolation

The cytoplasmic or nuclear location of *LOC441461* was assessed in MKN74 and SNU216 cells using a PARIS kit (Thermo Fisher Scientific, Waltham, MA, USA) according to the manufacturer’s protocol.

### 2.7. qRT-PCR

Total RNA was extracted from cells using an RNeasy mini kit (Qiagen, Hilden, Germany). The purity and concentration of total RNA were measured using a NanoDrop 1000 spectrophotometer (ND-1000, Thermo Fisher Scientific). One microgram of RNA was used for cDNA synthesis with an AccuPower CycleScript RT Premix (BIONEER) according to the manufacturer’s instructions. qPCR was performed on a CFX96 Touch real-time PCR detection system (Bio-Rad Laboratories, Hercules, CA, USA) using an iQ™ SYBR Green Supermix (Bio-Rad Laboratories) and the indicated primer sets (Appendix A). The following PCR reactions were performed: initial denaturation (95 °C for 10 min) followed by 45 cycles of denaturation (95 °C for 10 s) and annealing (55 °C for 30 s). Data were analyzed using the 2^−ΔΔCt^ method and normalized to GAPDH or U6 as an endogenous internal control.

### 2.8. Proliferation Assay

Living cells (5000 living cells/well) transfected with N.C or si-*LOC441461* were seeded in 96-well culture plates. After 6 h of transfection (or incubation) at 37 °C in 5% CO_2_ conditions, the medium was replaced with a fresh culture medium. Cell Counting Kit 8 solution (CCK8; Abcam, Cambridge, UK) was added directly to the cells. Cell growth was determined by measuring the absorbance at 450 nm every 24 h.

### 2.9. Colony Formation Assay

N.C- or si-*LOC441461*-transfected MKN74 cells were seeded into 6-well plates at a density of 2500 cells per well. After 2 weeks of incubation at 37 °C in 5% CO_2_, cells were fixed with 3.7% formaldehyde for 10 min and stained with 0.01% crystal violet solution in 10% ethanol for 20 min. Colony formation was quantified using 33% acetic acid and measured on a VICROR3 Multilabel Plate Reader (PerkinElmer, Waltham, MA, USA) at a wavelength of 590 nm.

### 2.10. Cell Cycle Assay

A total of 1 × 10^6^ cells were harvested and fixed with 70% ethanol added dropwise to the pellet. After 12 h of incubation at −20 °C, cells were stained with PI/RNase Staining Buffer (BD Biosciences, San Jose, CA, USA) for 15 min at RT and detected using flow cytometry (Beckman Coulter, Brea, CA, USA). Cell cycle progression was analyzed using FlowJo v. 10.8.1 (FlowJo LLC, Ashland, OR, USA) with the Watson (Pragmatic) model.

### 2.11. Apoptosis Assay

N.C- or si-*LOC441461*-transfected MKN74 and SNU216 cells were seeded into 6-well plates at a density of 3 × 10^5^ cells per well and incubated in the presence of 5-FU (2.5 μg/mL). After 24 h, apoptotic cells were detected with an Annexin V Apoptosis Detection Kit (BD Biosciences), following the manufacturer’s instructions using flow cytometry (Beckman Coulter).

### 2.12. Migration and Invasion Assays

Migration and invasion assays were performed using Transwell plates (Corning, NY, USA). N.C- or si-*LOC441461*-transfected MKN74 and SNU216 cells (7.5 × 10^4^) were seeded into the upper chamber of Transwell plates in a serum-free medium. For invasion assays the upper chamber was precoated with Matrigel Matrix (Corning). After incubation under culture conditions for 24 h, the lower chamber of the Transwell plates was fixed with 10% formaldehyde solution, and cells on the upper chamber were removed with cotton swabs. A 0.01% crystal violet solution was used to stain cells, to detect migration and invasion. After air-drying, the stained solution was eluted using 33% acetic acid, and the absorbance of the eluted solution was measured using a VICROR3 Multilabel Plate Reader (PerkinElmer) at a wavelength of 590 nm.

### 2.13. Wound Healing Assay

For the wound healing assay, a 24-well culture insert dish (ibid GmbH, Münster, Germany) was used to determine the migration ability of MKN74 cells. N.C- and si-*LOC441461*-transfected MKN74 cells were seeded onto the culture inserts and incubated for 12 h after a confluent monolayer was formed. The inserts were gently removed, and a complete growth medium was added. Wound closure at 24 and 48 h was photographed under a microscope. The wound area was measured and quantified using ImageJ v. 1.53k (National Institutes of Health, Bethesda, MD, USA).

### 2.14. RNA-Seq Analysis

The total RNA of N.C- or si-*LOC441461*-transfected MKN74 cells was extracted with Trizol (Invitrogen), and the concentration was calculated using Quant-IT RiboGreen (Invitrogen). The integrity of the total RNA was measured on TapeStation RNA ScreenTape (Agilent Technologies, Santa Clara, CA, USA), and an RNA integrity number >7.0 was used for RNA library construction.

Total RNA (0.5 µg) was used to construct a library using an Illumina TruSeq Stranded Total RNA Library Prep Gold Kit (Illumina, Inc., San Diego, CA, USA). After removing the rRNA, fragmentation was performed with mRNA under elevated temperatures. First-strand cDNA was synthesized using SuperScript II Reverse Transcriptase (Invitrogen) and random primers. Second-strand cDNA synthesis was performed using DNA polymerase I, RNase H, and dUTP. These cDNA fragments went through an end repair process mediated by the addition of a single ‘A’ base, as well as the ligation of the adapters. The products were purified and enriched by PCR to create the final cDNA library.

The libraries were quantified using KAPA Library Quantification Kits for Illumina sequencing platforms according to the qPCR Quantification Protocol Guide (KAPA BIOSYSTEMS, #KK4854) and qualified using the TapeStation D1000 ScreenTape (Agilent Technologies, #5067-5582). Indexed libraries were then submitted to Illumina NovaSeq (Illumina), and paired-end (2 × 100 bp) sequencing was performed by Macrogen Incorporated (Seoul, Republic of Korea). Data have been deposited in the Gene Expression Omnibus (http://www.ncbi.nlm.nih.gov/geo/ accessed on 8 January 2022), under the accession number GSE193700.

### 2.15. Differentially Expressed Gene Analysis

Differentially expressed genes between N.C and si-*LOC441461* groups were analyzed using the DESeq2 package, which is included in R [25]. Genes with an absolute value of log2 fold change > 1 and a Benjamini–Hochberg-adjusted *p*-value < 0.05 were included.

### 2.16. Pathway Enrichment Analysis

Reactome pathway (https://reactome.org/ accessed on 18 November 2021) enrichment analysis was performed using ShinyGO software, and significant pathways with an FDR correction < 0.05 were extracted in ShinyGO v. 0.741 [26]. The *p*-values of the enriched pathways are presented as −log10 (*p*-value).

### 2.17. Transcription Factor Enrichment Analysis Using the LncRNA-TF Interactome

Molecules interacting with *LOC441461* were predicted using RNAinter v. 4.0 [27]. Transcription factor enrichment analysis was performed to identify factors affected by *LOC441461* using ChEA (https://www.encodeproject.org/chip-seq/ accessed on 18 November 2021) and ENCODE ChIP-seq databases (https://maayanlab.cloud/chea3/ accessed on 18 November 2021), and the differentially expressed gene set between N.C and *LOC441461* knockdown samples using ShinyGO [26]. The common transcription factors were visualized using Venny v. 2.1.0 (https://bioinfogp.cnb.csic.es/tools/venny/ accessed on 20 November 2021).

### 2.18. Transcription Factor Target Gene Extraction

The target genes of each transcription factor were extracted according to cotarget genes from ENCODE ChIP-seq data, based on the target gene database [28] and the ChEA ChIP-X target gene database [29].

### 2.19. Hierarchical Clustering

Hierarchical clustering was performed to examine the differences in the expression of target genes between the si-*LOC441461* and N.C groups. The expression level of genes was normalized to transcripts per million. ComplexHeatmap v. 2.8.0, a package of R, was used to visualize the heatmap with annotation group information [30].

### 2.20. Statistical Analysis

Significant differences between treatment means were determined using Student’s *t*-test. All qRT-PCR, cell proliferation, colony formation, cell cycle, apoptosis, migration, invasion, wound healing, and RNA-seq assays were performed in triplicate. Error bars indicate standard deviation. One asterisk (*) denotes *p* < 0.05; two asterisks (**) denote *p* < 0.01; three asterisks (***) denote *p* < 0.001; and four asterisks (****) denote *p* < 0.0001.

## 3. Results

### 3.1. Comparison of Gene Expression between Stage I and Stage IV Gastric Cancer

The result of IG feature selection between TNM stage I and IV gastric cancer patients using RNA-seq data identified 59 genes with an IG value > 0. The 59 genes were subjected to a *t*-test comparing TNM stage I and stage IV, and seven statistically significant genes were identified after Bonferroni correction. The gene names, including IG scores and *p*-values determined via a t-test, are presented in Table 1. The IG value results are presented in Appendix A. The *p*-values of genes with an IG >0 determined by t-test are presented in Appendix A.

### 3.2. Comparison of LOC441461 Expression and Prognosis between Different Gastric Cancer Stages

The expression of *LOC441461* was assessed in samples from different TNM stages and compared between tumor and normal tissues. *LOC441461* expression was lower in stage IV than in stages I, II, and III, whereas its expression was similar between stages I, II, and III (Figure 1A). *LOC441461* expression did not differ between gastric cancer and adjacent normal tissues (Figure 1B). As *LOC441461* expression differed between stage IV and stage I, II, and III gastric cancer, a logistic regression analysis was performed according to the T, N, and M categories. T4 tumors showed lower *LOC441461* expression than T1, T2, and T3 tumors, whereas there was no difference between the different N categories. In the M category, *LOC441461* expression was lower in M1 than in M0 tumors (Table 2). Based on the difference in expression of *LOC441461* in the T and M categories, a survival analysis was conducted with overall survival (OS) and progression-free interval (PFI). The combination of the T category and the M category was divided into six groups. The difference in the prognosis in these six groups was significant in both OS and PFI (Figure 1C,D). The prognoses of T4 with M0 and M1 with T4 were significantly worse than T1 with M0 and M0 with T4, respectively, in OS (Appendix A). Otherwise, only T4 with M0 showed a worse prognosis than T1 with M0 in PFI (Appendix A). The entire results of the logrank test are described in Appendix A.

### 3.3. Selection of Gastric Cancer Cell Lines and si-LOC441461 Treatment 

The effect of *LOC441461* on gastric cancer cells was investigated based on the TCGA STAD data. *LOC441461* expression was first measured in different gastric cancer cell lines, to select the most adequate cells. The MKN74 and SNU216 cell lines showed high expression of *LOC441461* and were, thus, selected for further experiments (Figure 2A). MKN74 and SNU216 cells were transfected with si-*LOC441461*, and depletion of *LOC441461* was detected (Figure 2B). *LOC441461* expression was higher in the nucleus than in the cytoplasm in both the SNU216 and MKN74 cell lines (Figure 2C).

### 3.4. Downregulation of LOC441461 Induces Proliferation and Accelerates Cell Cycle Progression

Lower *LOC441461* expression was correlated with increased tumor volume, suggesting that negative *LOC441461* expression affects the proliferation ability of MKN74 and SNU216 cells. The results of the CCK8 assay showed that *LOC441461* significantly inhibited the proliferation of MKN74 and SNU216 cells, suggesting that *LOC441461* promotes gastric cancer cell proliferation in vitro (Figure 3A and Appendix A). *LOC441461* knockdown induced cell cycle progression; specifically, the G1/S transition was accelerated (Figure 3B). However, the colony formation ability, which indicates the ability of a single cell to grow into a colony, was significantly decreased in response to si-*LOC441461* in both MKN74 and SNU216 cells (Figure 3C). 

### 3.5. LOC441461 Knockdown Promotes 5-FU-Induced Gastric Cancer Cell Apoptosis

The present results indicated that *LOC441461* knockdown induced gastric cancer cell apoptosis in vitro. As shown in Figure 4, *LOC441461* knockdown promoted apoptosis in MKN74 and SNU216 cells treated with 5-FU (Figure 4A,B). We, thus, examined the effect of *LOC441461* on apoptosis and drug resistance in MKN74 and SNU216 cells treated with 5-FU. The results showed that the expression of *LOC441461* correlated negatively with the drug sensitivity of gastric cancer (Figure 4A); whereas, DMSO treatment decreased the rate of apoptosis of *LOC441461* knockdown cells (Figure 4A,B).

### 3.6. LOC441461 Knockdown Increases Gastric Cancer Cell Motility and Invasiveness

Analysis of the migration and invasion of gastric cancer cells using the Transwell assay showed that SNU216 cells had a higher rate of migration and invasiveness than MKN74 cells (Appendix A). The knockdown of *LOC441461* increased the migration and invasion capability of SNU216 cells (Figure 5A). Additionally, the migration and invasion abilities of SNU216 cells were also significantly increased in the absence of *LOC441461* (Figure 5B). The results of the wound healing assay showed that wound closure was significantly higher in *LOC441461*-silenced MKN74 cells than in the negative control group (Figure 5C). The decrease in the wound area was statistically significant after 48 h, but not at 24 h.

### 3.7. Effect of LOC441461 Knockdown on the Transcriptomic Landscape of Gastric Cancer Cells

The transcriptomic landscape of *LOC441461* knockdown gastric cancer cells was analyzed by RNA sequencing. Three replicates were, respectively, included in two groups, including negative controls and the knockdown of *LOC441461*. Differentially expressed gene analysis was performed using expression data obtained by RNA-seq. A total of 2503 genes with an absolute log2 fold change >1 and an adjusted *p* < 0.05 were selected for pathway enrichment analysis. Of these, 1202 upregulated genes and 1301 downregulated genes in the knockdown group were selected. The genes are listed in Appendix A. The selected genes were subjected to pathway enrichment analysis using the Reactome pathway database. Apoptosis- and cell-cycle-progression-associated terms were significantly enriched in genes that were upregulated in the *LOC441461* knockdown group (Figure 6A). The detailed results are presented in Appendix A. The mRNA expression of cyclin D1, which is involved in the G1/S transition of the cell cycle, and TRAIL, which is associated with apoptosis and the epithelial-to-mesenchymal transition (EMT), were higher in the *LOC441461* knockdown group (Figure 6B).

### 3.8. LOC441461 Changes the Nature of Gastric Cancer Cells by Modulating Transcription Factor Activity

Analysis of the RNAinter database showed that most of the molecules interacting with *LOC441461* were transcription factors (Appendix A). To identify the transcription factors modulated by *LOC441461*, a transcription factor enrichment analysis was performed according to the ENCODE and ChEA consensus transcription factors obtained from the ChIP-seq database, as well as the differentially expressed genes from RNA-seq data (Appendix A). RELA and IRF1 interacted with *LOC441461* and were significantly enriched among the genes upregulated by *LOC441461* knockdown, whereas ESR1, AR, POU5F1, YY1, FOXM1, TRIM28, SMAD4, E2F4, and GATA1 interacted with *LOC441461* and were significantly enriched in the downregulated gene set (Figure 7A). Hierarchical clustering was performed based on the target gene expression of significantly enriched transcription factors from RNA-seq data. Transcription factors significantly affected by *LOC441461* were identified by hierarchical clustering, which showed that the expression of the RELA, IRF1, ESR1, AR, POU5F1, TRIM28, and GATA1 target genes differed significantly between the knockdown and negative control groups (Figure 7B). The detailed results of hierarchical clustering and the expression of each target gene are presented in Appendix A. Target genes were extracted by the cotarget genes in the ENCODE ChIP-seq database and ChEA ChIP-X database, except AR and SMAD4, which were extracted from the ChEA ChIP-X database. The target genes of each transcription factor are listed in Appendix A.

## 4. Discussion

Recent reports on the different functions of lncRNAs have highlighted their involvement in tumor development and progression [18,31]. Moreover, the investigation of the molecular and cellular functions of lncRNAs may lead to the identification of therapeutic targets and the design of strategies for the treatment of gastric cancer [18,31,32,33]. By examining the association between lncRNA *LOC441461* expression level and TNM stage, we identified biological processes in gastric cancer. In this study, *LOC441461* expression was lower in stage IV than in stage I, II, and III gastric cancer samples from TCGA STAD data. The association between the degree of the malignancy of gastric cancer and higher stages, as assessed by the TNM staging system, supports the effect of *LOC441461* downregulation on increasing the severity of the disease by promoting tumor growth, cell motility, and worse prognoses. Furthermore, the localization and local interactions of lncRNAs are key to predicting their function [34]. Overall, we suggest that *LOC441461* modulates gene transcription, inducing the malignancy of gastric cancer by interacting with RELA, IRF1, ESR1, AR, POU5F1, TRIM28, and GATA1.

LncRNAs are involved in diverse biological processes, such as proliferation, motility, and the EMT in different aspects of regulation [11,35]. The latest research shows that *LOC441461* upregulation promotes growth and motility in colon cancer cell lines [22]. In the present study we found that the different effects of *LOC441461* may be attributed to the use of gastric cancer, and several experimental validations were conducted to investigate the function of *LOC441461* in gastric cancer cells. The results of cell experiments and RNA-seq analysis showed that the downregulation of the lncRNA *LOC441461* could promote the growth and metastasis of gastric cancer cells in vitro and in silico. Furthermore, lncRNAs modulate the cell cycle by regulating the expression of cyclins and cyclin-dependent kinases [36,37], and a similar mechanism may underlie the regulation of the cell cycle by *LOC441461*. The FOXO-mediated transcription of cell cycle genes was significantly increased in *LOC441461* knockdown cells. However, the clonogenic ability showed a positive correlation with the expression of *LOC441461*.

5-FU treatment, a fluoropyrimidine chemotherapy, is the first treatment of choice of gastric cancer [9]. Previous studies suggest that 5-FU sensitivity increases the status of DNA damage [38,39]. Ultimately, the p53-independent DNA damage response was significantly enriched in *LOC441461* knockdown cells, suggesting that the DNA damage response was promoted by the effect of *LOC441461* knockdown, in inducing the 5-FU-induced apoptosis of gastric cancer cells.

In addition, it was found that lncRNAs regulate gastric cancer cells via diverse contexts, such as regulating the EMT process and PI3K/AKT pathway [22,31,40]. The EMT is considered as a key role in human malignancies activated during tumor metastasis [41]. Given the abovementioned findings, the results suggested that the knockdown of *LOC441461* increased the motility of gastric cancer cells. The downregulation of *LOC441461* increased the proportion of migrating/invading cells and promoted wound closure. The upregulation of tumor-necrosis-factor-related apoptosis-inducing ligand (TRAIL) signaling also promotes the EMT in various cancers [42,43,44,45]. By contrast, the interaction of RUNX1 with cofactors was significantly decreased in the absence of *LOC441461*, and RUNX1 is a suppressor of the EMT [46,47]. Thus, the EMT was promoted in relation to the depletion of *LOC441461*, as indicated by TRAIL signaling and RUNX1 interactions.

Multiple studies have observed that lncRNAs are, overall, more numerous in the nucleus than their cytoplasm [16,31,34,35,48]. Interestingly, the instability of nuclear lncRNAs can act as an oncogene or tumor suppressor and modulate transcription factors, which are key players in transcriptional regulation [16,31,48]. We investigated the potential mechanism underlying the modulation of gene expression by transcription factors that interact with *LOC441461* in gastric cancer cells. Transcription factor enrichment analysis was performed to identify transcription factors that interact with *LOC441461*. Hierarchical clustering based on the expression of the target genes of 11 transcription factors affected by *LOC441461* resulted in the classification of the target genes of seven transcription factors, RELA, IRF1, ESR1, AR, POU5F1, TRIM28, and GATA1, into two groups (Appendix A). Hierarchical clustering of the target genes of seven transcription factors also clearly divided them into two groups, in correlation with the expression of *LOC441461*. RELA, also called p65, is associated with NF-κB heterodimer formation, nuclear translocation, and activation [49]. The dysregulation of NF-κB/RELA promotes distant metastasis in gastric cancer, suggesting its potential as a novel therapeutic target [50]. IRF1 not only acts as a viral infection response transcription factor [51], but also plays a role in cell cycle arrest, enhancing 5-FU sensitivity and suppressing the EMT in gastric cancer [52,53,54]. In the absence of *LOC441461*, IRF1 expression was upregulated (Appendix A). ESR1 and androgen receptor (AR) upregulation is associated with a poor prognosis in gastric cancer patients [55]. POU5F1, a POU class 5 homeobox 1 member, is associated with a poor prognosis, and its expression is significantly increased after chemotherapy in colorectal cancer [56,57]. Tripartite motif-containing 28 (TRIM28) is associated with a poor prognosis by promoting tumor progression and the activation of autophagy in glioma [58,59]. GATA1 or GATA-binding factor 1 promote the EMT in breast cancer [60].

According to the results of an expression analysis between TNM stages and a survival analysis, the expression of *LOC441461* was associated with malignant characteristics of gastric cancer cells, including cell growth and motility. Specifically, the molecular function of *LOC441461* was bioinformatically predicted by modulating the activity of seven transcription factors, including RELA, IRF1, ESR1, AR, POU5F1, TRIM28, and GATA1. Based on these facts, they can be applied to the novel treatment management of gastric cancer in various ways. First, upregulating the expression of *LOC441461* in gastric cancer patients can suppress progression to a malignant status of gastric cancer. Second, modulating the activity of transcription factors interacting with *LOC441461* could be a candidate therapeutic target to manage advanced gastric cancer. Recently, RNA biopharmaceuticals have been attracting attention as a breakthrough clinical application of the COVID-19 RNA vaccine [33,61]. Proving its application as a gastric cancer treatment by regulating the expression of *LOC441461* will be a very interesting study in the future.

## 5. Conclusions

In this study, we identified the lncRNA *LOC441461* as a potential biomarker in patients with an advanced TNM stage, which has significant value in tumor development and progression. *LOC441461* knockdown promoted growth, cell cycle progression, motility, and invasiveness in gastric cancer cells in vitro. Additionally, molecular links between the expression of *LOC441461* and the nature of gastric cancer cells were investigated by measuring changes in gene expression modulated by transcription factors. Candidate transcription factors that may interact with *LOC441461* were identified using bioinformatic methods and various databases. Although this study has the limitation that the exact mechanism of action with which *LOC441461* interacted with seven transcription factors was only predicted by bioinformatical methods, and not validated by experimental methods, the findings in this study may lead to the development of novel therapeutic strategies for patients with gastric cancer.

## Figures and Tables

**Figure 1 cancers-14-01149-f001:**
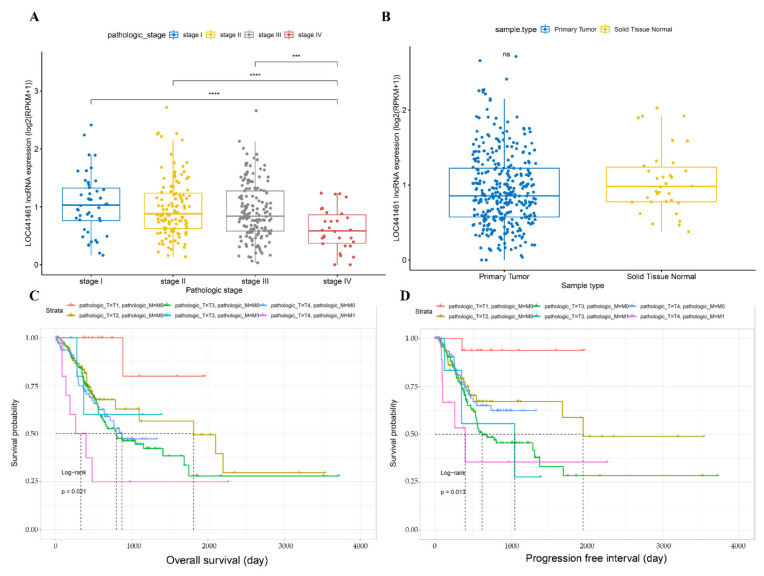
Comparison of *LOC441461* expression and survival analysis using TCGA STAD data. (**A**) The expression of *LOC441461* in each TNM stage and statistical analysis. The statistical analysis was conducted via a *t*-test. **** *p* < 0.0001, *** *p* < 0.001. (**B**) *LOC441461* expression in primary tumors and adjacent normal tissues. ns, not significant. (**C**,**D**) Comparison of prognoses between different combinations of T and M. The statistical method was a logrank test. Pathologic_T is the T category of TNM staging. Pathologic_M is the M category of TNM staging.

**Figure 2 cancers-14-01149-f002:**
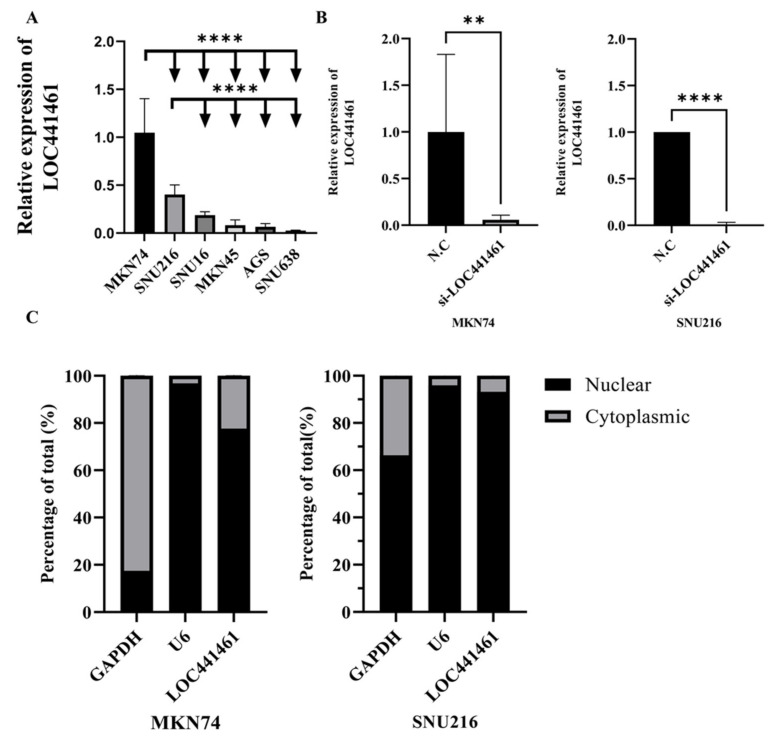
Relative expression of *LOC441461* and localization in human gastric cancer cell lines. (**A**) Expression levels of *LOC441461* were measured in six gastric cancer cell lines (MKN74, SNU216, SNU16, MKN45, AGS, and SNU638) by qRT-PCR. (**B**) Knockdown of *LOC441461* in MKN74 and SNU216 cells via transfection with small interfering RNA (siRNA) targeting *LOC441461* (si-*LOC441461*) compared with a scrambled negative control (N.C). qRT-PCR was performed to quantify the relative expression levels of *LOC441461* normalized to GAPDH; SNU638 cells were compared as a reference sample. (**C**) Subcellular fractionation was performed using a PARIS kit to separate the nuclear and cytoplasmic fractions of MKN74 and SNU216 cells. The localization of *LOC441461* in the nucleus (normalized to U6) and cytoplasm (normalized to GAPDH) was compared via qRT-PCR using the 2^−ΔΔCt^ method. All experiments were performed in triplicate, and data are expressed as the mean ± standard deviation (** *p* < 0.01, **** *p* < 0.0001; Student’s *t*-test).

**Figure 3 cancers-14-01149-f003:**
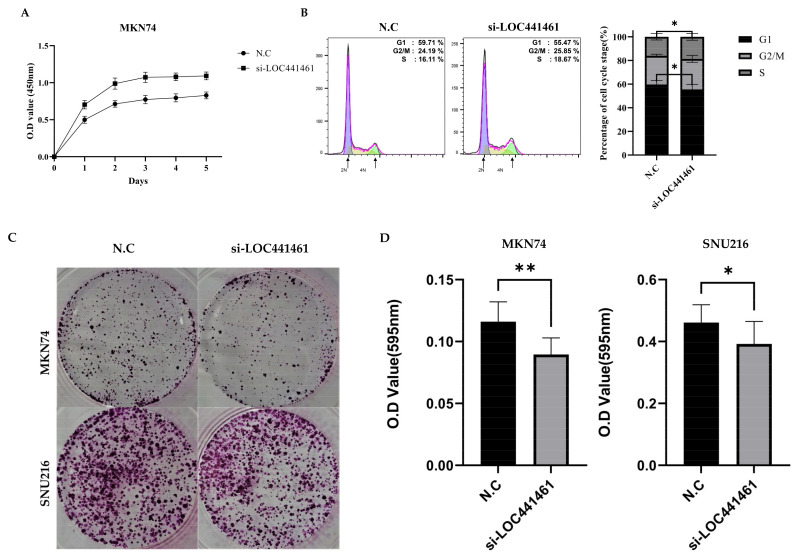
*LOC441461* suppresses cell proliferation by promoting G1/S transition in human gastric cancer cells. (**A**) Knockdown of *LOC441461* by transfection with si-*LOC441461* or a scrambled negative control (N.C) in MKN74 cells. The proliferation ability of N.C- and si-*LOC441461*-transfected cells was examined using the CCK-8 solution at every 24 h. (**B**) The cell cycle progression of N.C- and si-*LOC441461*-transfected cells was examined using flow cytometry with the PI/RNase solution. Each cell cycle phase was analyzed with FlowJo v. 10.8.1. Graph of each quantified phase. (**C**) MKN74 and SNU216 cells were transfected with N.C or si-*LOC441461* and stained with 0.01% crystal violet solution after 14 days. (**D**) Graph of colony formation ability after elution with 33% acetic acid. All experiments were performed in triplicate, and data are expressed as the mean ± standard deviation (* *p* < 0.05, ** *p* < 0.01; Student’s *t*-test).

**Figure 4 cancers-14-01149-f004:**
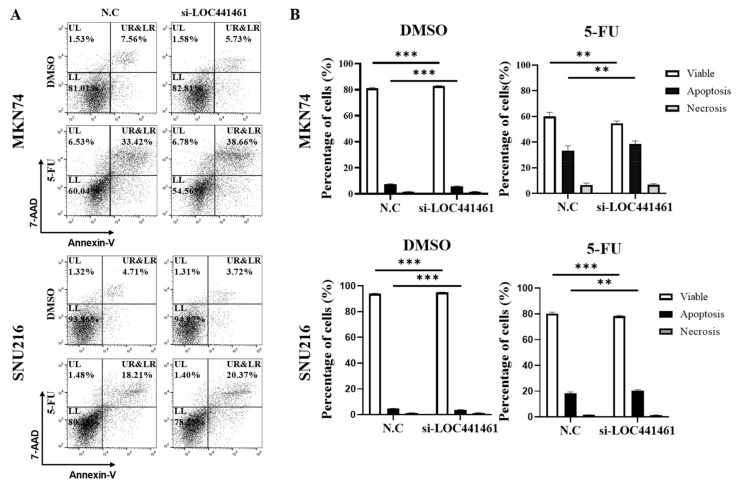
Sensitivity to 5-FU was increased in gastric cancer cells with *LOC441461* knockdown. MKN74 and SNU216 cells were transfected with a scrambled negative control (N.C) or si-*LOC441461*. After exposure to 5-FU (2 μg/mL), apoptotic cells were measured using an annexin V/7-AAD kit. (**A**) Representative dot plot of the apoptosis assay using flow cytometry. The upper-left (UL) quadrant indicates necrotic cells, the lower-left (LL) quadrant shows viable cells, the lower-right (LR) quadrant indicates early apoptotic cells, and the upper-right (UR) quadrant shows the late stage of apoptosis. (**B**) Graph showing the percentage of the cells in each stage. Experiments were performed in triplicate, and data are expressed as the mean ± standard deviation (** *p* < 0.01; *** *p* < 0.001; Student’s *t*-test).

**Figure 5 cancers-14-01149-f005:**
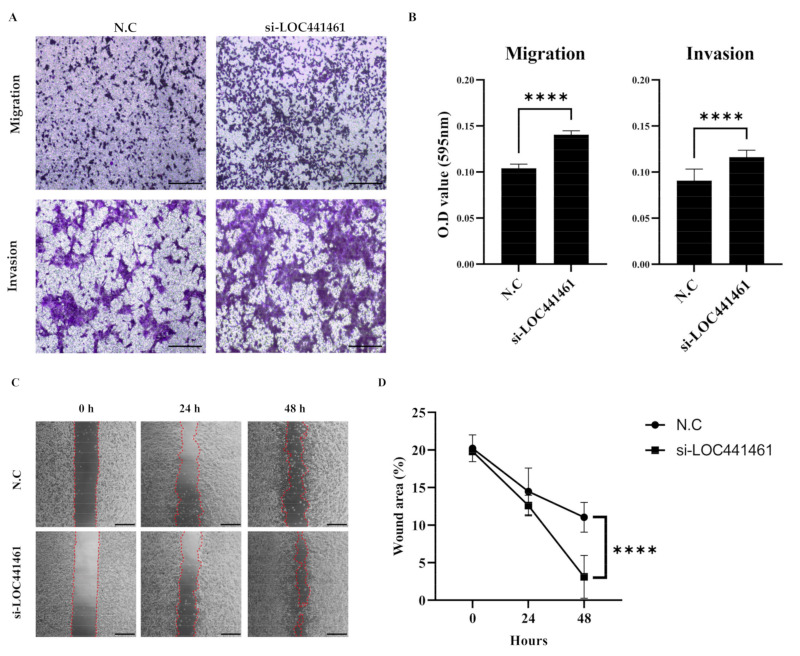
*LOC441461* knockdown promotes gastric cancer cell migration and invasion. (**A**) Representative microscopic images of the Transwell migration and invasion assays in SNU216 cells (stained with crystal violet). Scale bar = 200 μm. (**B**) Relative migration and invasion abilities were quantified after elution with 33% acetic acid. (**C**) Representative images of wound healing assays using culture preinserts in MKN74 cells with *LOC441461* knockdown. Scale bar = 500 μm. (**D**) Wound closure was assessed by ImageJ v. 1.53k from 0 to 48 h. All experiments were performed in triplicate, and data were expressed as the mean ± standard deviation (**** *p* < 0.0001; Student’s *t*-test).

**Figure 6 cancers-14-01149-f006:**
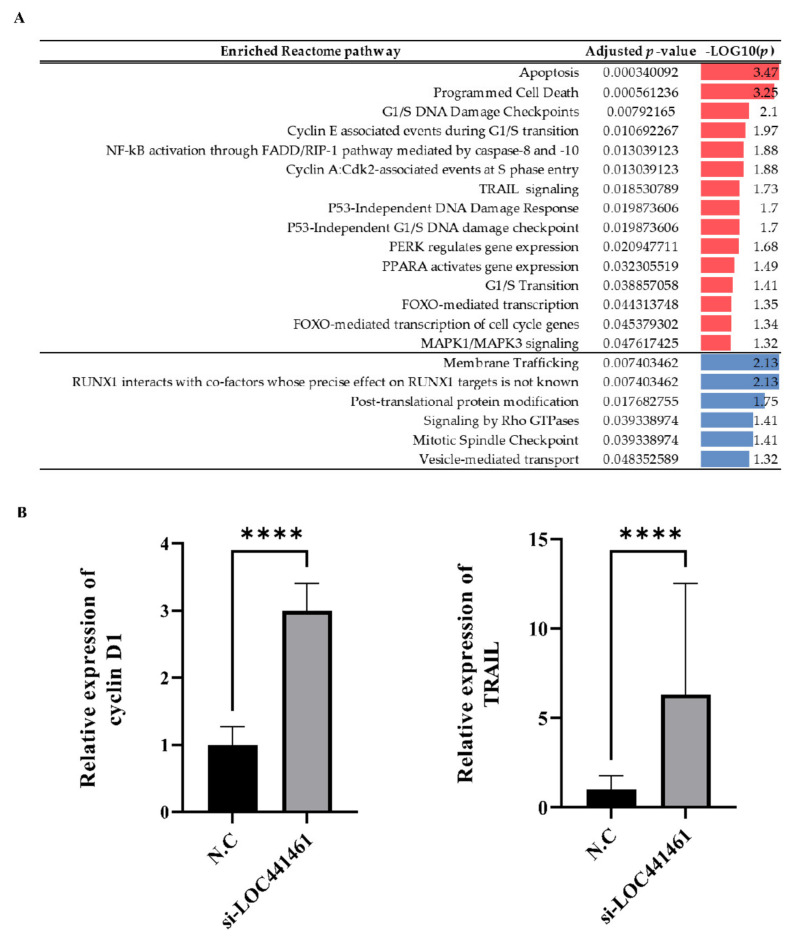
Transcriptomic landscape after the depletion of *LOC441461* in gastric cancer cell lines. (**A**) Results of pathway enrichment analysis. Red indicates upregulated pathways in *LOC441461* knockdown groups. Blue indicates downregulated pathways in *LOC441461* knockdown groups. (**B**) mRNA expression of cyclin D1 and TRAIL determined by qRT-PCR in MKN74 cells transfected with N.C or si-*LOC441461*. The data were normalized to GAPDH. All experiments were performed in triplicate, and data are expressed as the mean ± standard deviation (**** *p* < 0.0001; Student’s *t*-test).

**Figure 7 cancers-14-01149-f007:**
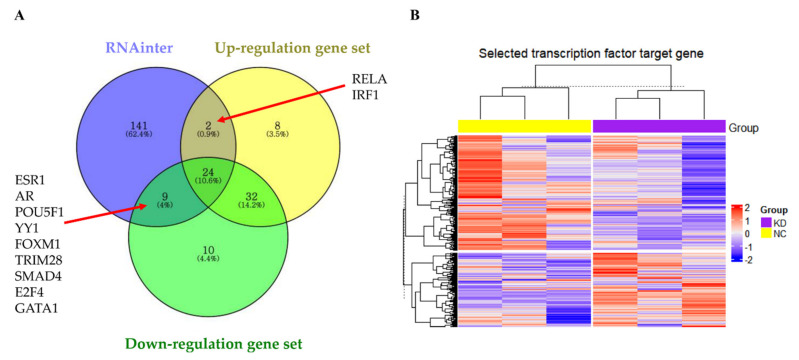
Investigation of the role of *LOC441461* in modulating transcription factor activity. (**A**) Enriched transcription factors in the up- or downregulated gene sets in interacting with *LOC441461*. (**B**) Hierarchical clustering based on the target gene expression of 11 transcription factors with RNA-seq data of *LOC441461* knockdown. KD, knockdown of *LOC441461*. N.C, negative control.

**Table 1 cancers-14-01149-t001:** Selected genes with IG and *p*-values according to *t*-tests based on XENA TCGA STAD.

Gene	IG	*p*
*NOV*	0.1150726	9.48 × 10^−6^
*CPA3*	0.1117299	4.84 × 10^−5^
*CARD11*	0.1049524	5.29 ×10^−5^
*ZNF467*	0.1327168	7.48 × 10^−5^
*AK097446*	0.1205477	7.52 × 10^−5^
*LOC441461*	0.1223392	8.80× 10^−5^
*TPSAB1*	0.1262245	0.0001181

IG, information gain. *p*, *p*-value of *t*-test.

**Table 2 cancers-14-01149-t002:** Results of the logistic regression analysis of TNM scores based on XENA TCGA STAD data.

TNM	Odds Ratio	CI—Low	CI—High	*p*
T1 vs. T2	0.30	0.09	0.88	0.086
T1 vs. T3	0.46	0.20	1.11	0.141
T1 vs. T4	0.1	0.02	0.32	0.001 *
T2 vs. T3	1.21	0.70	2.14	0.677
T2 vs. T4	0.42	0.20	0.87	0.015 *
T3 vs. T4	0.38	0.21	0.67	0.002 *
N0 vs. N1	0.84	0.48	1.45	0.68
N0 vs. N2	0.90	0.50	1.60	0.777
N0 vs. N3	0.58	0.30	1.06	0.047
N1 vs. N2	1.08	0.56	2.08	0.508
N1 vs. N3	0.65	0.32	1.29	0.089
N2 vs. N3	0.63	0.30	1.26	0.037
M0 vs. M1	0.08	0.02	0.3	0.001 *

CI, confidence interval. *p*, *p*-value determined via logistic regression. * Statistically significant *p*-value from logistic regression analysis after FDR correction.

## Data Availability

RNA-seq data have been deposited in the Gene Expression Omnibus (http://www.ncbi.nlm.nih.gov/geo/ accessed on 18 November 2021) under the accession number GSE193700 (https://www.ncbi.nlm.nih.gov/geo/query/acc.cgi?acc=GSE193700 accessed on 18 November 2021).

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
