# Peer review of "Downregulation of *LOC441461* Promotes Cell Growth and Motility in Human Gastric Cancer"

_cancers, 2022, doi:10.3390/cancers14051149_

Round 1
Reviewer 1 Report
This study is interesting. The authors demonstrate the role of LOC441461 in gastric cancer cells. The summary is LOC441461 may be as a potential biomarkers therapeutic targes for the treatment of gastric cancer. But there are some questions in this study.
- Is the topic "Downregulation of long non-coding RNA LOC441461 STX-antisense 1 RNA promotes cell growth and motility in gastric cancer cell" or "Downregulation of LOC441461 promotes cell growth and motility in human gastric cancer"?
- In Figure 2B, what is the standard to be used to normalize?
- In Figure 3A, what is the culture condition at day3-5 (after transfection), keeping the transfection reagent and siRNA or changing new culture medium?
- In Figure 4, the cell numbers for flow cytometry are different between DMSO group and 5-fu group. If the cell number from all groups are the same, the result is not significant.
- Most of cell numbers are wrong. Line 117, 124, and 129, 1x106, 3x105, and 7.5x104 are wrong.
- Line 107-108, cell growth was determined by measuring absorbance at 460 nm every 24 h. However, the data showed 450nm.
- Line 113, Colony formation was quantified using 10% acetic acid. However, the figure legend is 33%. Which one is current?
- In Figure 3A (Line 258-259), the proliferation ability of N.C and si-LOC441461-transfected cells was examined using the CCK-8 solution at 24 and 48 h. However, the data showed the proliferation from day 1 to day 5.
- Why did the data of proliferation, cell cycle, wound healing assays, and combination with 5-fu in SNU216 cells not show in the manuscript or supplementary data?
Author Response
Response to Reviewer 1 Comments
ANSWERS TO THE REVIEWERS' COMMENTS
We thank you and the reviewers for the valuable comments on our manuscript. Our responses to the comments are provided below. The comments have substantially improved the quality of the manuscript. The revisions in the manuscript are marked up using the “Track Changes” function, and the relevant line numbers are provided with our response. The response of “Autor’s Reply to the Review Report (Reviewer 1)” has been written in red as an example.
Review Comments to the Author:
Reviewer #1: This study is interesting. The authors demonstrate the role of LOC441461 in gastric cancer cells. The summary is LOC441461 may be as a potential biomarkers therapeutic targets for the treatment of gastric cancer. But there are some questions in this study.
Comment 1
|
Is the topic "Downregulation of long non-coding RNA LOC441461 STX-antisense 1 RNA promotes cell growth and motility in gastric cancer cell" or "Downregulation of LOC441461 promotes cell growth and motility in human gastric cancer"? |
Response: Thank you for pointing out our critical mistake. The topic of this study is "Downregulation of LOC441461 promotes cell growth and motility in human gastric cancer".
Comment 2
|
In Figure 2B, what is the standard to be used to normalize? |
Response: In response to the reviewers’ valuable comments, we additionally describe the normalization method in Figure 2.
Comment 3
|
In Figure 3A, what is the culture condition at day3-5 (after transfection), keeping the transfection reagent and siRNA or changing new culture medium? |
Response: Thanks for your critical comments. We incubated the transfected cells with medium containing transfection reagent and siRNA for 6 h, then replaced with fresh culture medium (excluding transfection reagent and siRNA). The modifications are described in lines 149-150 of the revised manuscript ('Proliferation assay' in the Materials and Methods section).
Comment 4
|
In Figure 4, the cell numbers for flow cytometry are different between DMSO group and 5-fu group. If the cell number from all groups are the same, the result is not significant. |
Response: We appreciate your critical comments. The number of cells has been changed, along with the relative percentage (%) in each status of cells. Figure 4 has been corrected and the modifications have been described in lines 356-364 of the revised manuscript.
Comment 5
|
Most of cell numbers are wrong. Line 117, 124, and 129, 1x106, 3x105, and 7.5x104 are wrong. |
Response: Thank you for pointing out our critical mistake. The mentioned typo has been corrected on lines 116, 132, and 140.
Comment 6
|
Line 107-108, cell growth was determined by measuring absorbance at 460 nm every 24 h. However, the data showed 450nm. |
Response: Thank you for pointing out our mistake. As shown in Figure 3A, the absorbance was measured at 450 nm. Corresponding line 152 has been corrected.
Comment 7
|
Line 113, Colony formation was quantified using 10% acetic acid. However, the figure legend is 33%. Which one is current? |
Response: Thanks for pointing out the major mistake. Line 157 has been corrected. In our study, cells stained in the colony formation assay were eluted with 33% acetic acid.
Comment 8
|
In Figure 3A (Line 258-259), the proliferation ability of N.C and si-LOC441461-transfected cells was examined using the CCK-8 solution at 24 and 48 h. However, the data showed the proliferation from day 1 to day 5. |
Response: Thank you for pointing out critical mistake. Proliferation ability was measured every 24 hours until day 5. Figure 3A legend has been corrected on line 339.
Comment 9
|
Why did the data of proliferation, cell cycle, wound healing assays, and combination with 5-fu in SNU216 cells not show in the manuscript or supplementary data? |
Response: Thank you to the reviewer for pointing out. The apoptosis result and proliferation result using SNU216 cells have been added to Figure 4 and Figure S1, respectively.
The wound healing assay results using SNU216 cells were not reproducible. The reason seems to be that the SNU216 cell line has half/semi-adhesive/suspension properties. Some cells float in suspension and interfere with the wound area during wound healing assays.
In addition, the quality of cell cycle data can be assessed by looking at the coefficient of variation (CV) of the G1 peak. A lower CV indicates better data quality, but a CV of less than 6% generally indicates acceptable data. SNU216 cell cycle analysis repeatedly shows very high CV values. Overall, please refer to the attached photos (culture morphology) of Korean Cell Line Bank (KCLB; Seoul,
Republic of Korea) that purchased SNU216 cell line.

Reviewer 2 Report
I have only some minor revisions to reccomend:
1. The authors did not specify the number of replicates in RNA-Seq analysis
2. Since the data of TCGA have been used for the analysis, TGCA should be inserted in acknowlegment following TGCA guidelines on their website
3. Each figure/table should report in the legends when TGCA data are used (as table 1 and table 2 where this specification is missing)
4. Reference sample used in qPCR with relative expression analysis should be clearly indicated. Particulary in Figure 2A where on axis y the relative expression is indicate, it is not clear relative to what control, what reference sample. In this case, it is appropriate if authors analyze normaloid gastric cells to be used ad reference sample. In this bargraph also the significance of variation is not indicated
Author Response
Response to Reviewer 2 Comments
ANSWERS TO THE REVIEWERS' COMMENTS
We thank you and the reviewers for the valuable comments on our manuscript. Our responses to the comments are provided below. The comments have substantially improved the quality of the manuscript. The revisions in the manuscript are marked up using the “Track Changes” function, and the relevant line numbers are provided with our response. The response of “Autor’s Reply to the Review Report (Reviewer 2)” has been written in red as an example.
Reviewer #2: I have only some minor revisions to recommend.
Comment 1
|
The authors did not specify the number of replicates in RNA-Seq analysis |
Response: Thank you for helpful advice. Three samples in each two groups including negative control groups and knockdown of LOC441461 were used in conducting RNA-seq analysis. As reviewer mentioned, the number of replicates has been described in lines 228, and 358-359.
Comment 2
|
Since the data of TCGA have been used for the analysis, TGCA should be inserted in acknowlegment following TGCA guidelines on their website |
Response: As reviewer commented, acknowledgment of TCGA has been added in lines 546-547.
Comment 3
|
Each figure/table should report in the legends when TGCA data are used (as table 1 and table 2 where this specification is missing) |
Response: As reviewer mentioned, the specification of TCGA data has added to the legend in Table 1 on line 241 and Table 2 on line 262, respectively.
Comment 4
|
Reference sample used in qPCR with relative expression analysis should be clearly indicated. Particulary in Figure 2A where on axis y the relative expression is indicate, it is not clear relative to what control, what reference sample. In this case, it is appropriate if authors analyze normaloid gastric cells to be used ad reference sample. In this bargraph also the significance of variation is not indicated. |
Response: Thank you for helpful advice. Clinical data, survival data, and RNA-seq data showed that genes co-expressed with LOC441461 were significantly involved in tumor development. To explore the role of LOC441461 in gastric cancer, we firstly measured its expression levels by using qRT-PCR in 6 gastric cancer cell lines. The LOC441461 expression level was normalized with GAPDH, and cells with the lowest expression of LOC441461 were compared as a reference sample (SNU638). We discussed that for functional studies with dramatic effects, the relative expression level of LOC441461 would be displayed in the high group. So a series of experiments were performed using MKN74 or SNU216 cells. This havs been added in lines 288-289. In addition we have added significance on a bar graph in Figure 2A.

Reviewer 3 Report
In the present manuscript "Downregulation of LOC441461 promotes cell growth and motility in human gastric cancer" the authors have identified lncRNA LOC441461 as potential biomarker and therapeutic target for the treatment of gastric cancer. This is an interesting and significant study. However, there are a number of major issues that the authors have to address to be able to publish their results.
Major:
Introduction:
- Please provide more information on gastric cancer regarding the current clinical need on disease management.
- Please include more information on lncRNA mode of action.
Results:
- The authors should assess the clinical significance of LOC441461 regarding gastric cancer patients’ survival in a screening cohort or at least in a publicly available dataset (e.g. GEO dataset).
Discussion:
- The authors state that LOC441461 is a potentially important biomarker. Please discuss more the translational impact of your findings to support the statement.
- The authors should discuss more their findings with updated literature.
Minor:
Materials & Methods:
- Please correct typos:
“2.9. Cell cycle assay 116 A total of 1 × 106 cells were harvested and fixed with 70% ethanol added dropwise 117 to the pellet.”
“2.10. Apoptosis assay 122 N.C- or si-LOC441461-transfected cells were seeded into 6-well plates at a density of 123 3 × 105 cells per well and incubated in the presence of 5-FU (2.5 μg/mL).”
- Please include the statistical test performed for LOC441461 expression comparison in the TCGA STAD (Figure 1).
Results
- Please increase font size in figures 3-6. Additionally, please replace figure 1 with a sharper one.
Discussion
- Please include more details on the first paragraph of the discussion for the topic described.
Author Response
Response to Reviewer 2 Comments
ANSWERS TO THE REVIEWERS' COMMENTS
We thank you and the reviewers for the valuable comments on our manuscript. Our responses to the comments are provided below. The comments have substantially improved the quality of the manuscript. The revisions in the manuscript are marked up using the “Track Changes” function, and the relevant line numbers are provided with our response. The response of “Autor’s Reply to the Review Report (Reviewer 2)” has been written in red as an example.
Reviewer #2: In the present manuscript "Downregulation of LOC441461 promotes cell growth and motility in human gastric cancer" the authors have identified lncRNA LOC441461 as potential biomarker and therapeutic target for the treatment of gastric cancer. This is an interesting and significant study. However, there are a number of major issues that the authors have to address to be able to publish their results.
Major points
Introduction
Comment 1
|
Please provide more information on gastric cancer regarding the current clinical need on disease management. |
Response: We would like to thank the reviewer for the suggestion. GC epidemiology, characteristics, and current clinical needs were added to the introduction section (Lines 31-59) .
Comment 2
|
Please include more information on lncRNA mode of action. |
Response: Thank you for helpful advice. The mode of action of lncRNA was added to the introduction section by referring to the recent publication (Lines 60-84).
Results
Comment 3
|
The authors should assess the clinical significance of LOC441461 regarding gastric cancer patients’ survival in a screening cohort or at least in a publicly available dataset (e.g. GEO dataset). |
Response: Thank you for helpful advise about conducting survival analysis. Using TCGA databse, we conducted survival analysis with overall survival and progression free interval between the combination of T and M category which shows significant difference of expression in the results of logistic regression analysis (Table 2). The results showed significantly different prognosis between six groups, specifically, T1 and T4 with M0, M0 and M1 with T4 in overall survival. In progression free interval, T1 and T4 with M0 showed significant different prognosis. Survival analysis was visualized as Kaplan-Meior curve in Figure 1C and Figure 1D. These also have been described in results section, lines 286-293 and lines 297-301. Additional results of log-rank test have been added to Table S5.
Discussion
Comment 4
|
The authors state that LOC441461 is a potentially important biomarker. Please discuss more the translational impact of your findings to support the statement. |
Response: Thank you for helpful comments. As reviewer mentioned, we have added translational impact of our findings in discussion section.
Comment 5
|
The authors should discuss more their findings with updated literature. |
Response: Thank you for reviewers’ suggestion, we agree that our discussion part is focused on our results. To avoid any confusion to the reader, we have removed those statement from the manuscript. We have instead stated with current publications and additional impacts on our study.
Minor points
Materials & Methods
Comment 6
|
Please correct typos: “2.9. Cell cycle assay 116 A total of 1 × 106 cells were harvested and fixed with 70% ethanol added dropwise 117 to the pellet.”
“2.10. Apoptosis assay 122 N.C- or si-LOC441461-transfected cells were seeded into 6-well plates at a density of 123 3 × 105 cells per well and incubated in the presence of 5-FU (2.5 μg/mL).” |
Response: Thank you for pointing out our critical mistake. The typos mentioned have been corrected. Line 116; 1 × 106 → 1 × 106, line 132; 3 × 105 → 3 × 105.
Comment 7
|
Please include the statistical test performed for LOC441461 expression comparison in the TCGA STAD (Figure 1). |
Response: Thank you for helpful comment. As reviewer mentioned, we added the information of statistical test performed in Figure 1 on line 299-301.
Results
Comment 8
|
Please increase font size in figures 3-6. Additionally, please replace figure 1 with a sharper one. |
Response: Thank you for helpful advice. As reviewer commented, we have changed font size of figures 3-6 more bigger and replaced high resolution image for Figure 1.
Discussion
Comment 9
|
Please include more details on the first paragraph of the discussion for the topic described. |
Response: Thank you for helpful advice. The first paragraph of the discussion has been modified in the revised manuscript as suggested (Lines 437-450).

Reviewer 4 Report
In the present manuscript entitled “Downregulation of LOC441461 promotes cell growth and motility in human gastric cancer” the authors have reported that lncRNA LOC441461 is negatively correlated with advanced TNM stage and its downregulation induces proliferation, cell cycle progression, and metastasis of GC cells in vitro.
This is a well-designed study with sufficient methodology. The results are well presented and support the working hypothesis and the general conclusions of the manuscript.
However, there are a number of issues that the authors have to address, to be able to publish their results.
- The introduction section of Gastric Cancer (GC) epidemiology and characteristics is poor and not informative about gastric cancer’s current clinical needs. Please provide more information on GC regarding its subcategories as well as diagnosis, prognosis and therapeutic management.
- TNM staging is a globally recognized and established classification system. Thus, its definition in the introduction section is not necessary and should be omitted.
- In general, lncRNAs paragraph needs improvement.
- Please provide an accurate definition of lncRNAs and their most significant characteristics.
- The authors haven’t comprehensively presented the most important molecular functions of lncRNAs. For example: At the transcriptional level, lncRNAs can act as scaffold (guides) or as architects and enhancer-like lncRNAs, while at the post-transcriptional level, they play a very crucial role in mRNA processing.
- It is not clear why the authors chose LOC441461 for further examination compared to the other genes in Table 1.
- Which RNA-seq data units (RPKM?) were retrieved from xena browser? Please include it in the material and methods section as well as in Figure 1.
- In figure 1 please change mRNA to RNA as LOC441461 is lncRNA.
- Have you evaluated LOC441461 prognostic value concerning disease progression and overall survival in STAD TCGA dataset? It is highly suggested that the authors include a more extensive clinical analysis of LOC441461 using STAD TCGA dataset as well as other publicly available datasets (e.g GEO datasets) in order to strengthen its clinical significance.
- In general, the Discussion section needs improvement, since the authors mainly discuss their results. Please include current publications and discuss more the translational impact of your study.
- In the discussion section the authors should indicate the most significant limitations of their work.
Author Response
Response to Reviewer 3 Comments
ANSWERS TO THE REVIEWERS' COMMENTS
We thank you and the reviewers for the valuable comments on our manuscript. Our responses to the comments are provided below. The comments have substantially improved the quality of the manuscript. The revisions in the manuscript are marked up using the “Track Changes” function, and the relevant line numbers are provided with our response. The response of “Autor’s Reply to the Review Report (Reviewer 3)” has been written in red as an example.
Review Comments to the Author:
Reviewer #3: In the present manuscript entitled “Downregulation of LOC441461 promotes cell growth and motility in human gastric cancer” the authors have reported that lncRNA LOC441461 is negatively correlated with advanced TNM stage and its downregulation induces proliferation, cell cycle progression, and metastasis of GC cells in vitro.
This is a well-designed study with sufficient methodology. The results are well presented and support the working hypothesis and the general conclusions of the manuscript.
However, there are a number of issues that the authors have to address, to be able to publish their results.
Comment 1
|
The introduction section of Gastric Cancer (GC) epidemiology and characteristics is poor and not informative about gastric cancer’s current clinical needs. Please provide more information on GC regarding its subcategories as well as diagnosis, prognosis and therapeutic management. |
Response: We would like to thank the reviewer for the suggestion. GC epidemiology, characteristics, and current clinical needs were added to the introduction section (Lines 31-59) .
Comment 2
|
TNM staging is a globally recognized and established classification system. Thus, its definition in the introduction section is not necessary and should be omitted. |
Response: Thank you for giving important advice. As reviewer commented, we deleted the definition of TNM staging.
Comment 3
|
In general, lncRNAs paragraph needs improvement. |
Response: Thank you for helpful advice. The revision of comment 3 is presented below.
Comment 3-1
|
Please provide an accurate definition of lncRNAs and their most significant characteristics. |
Response: Thank you for suggesting meaningful advice. As reviewer commented, we added accurate definition of lncRNAs and their most significant characteristics (Lines 60-84).
Comment 3-2
|
The authors haven’t comprehensively presented the most important molecular functions of lncRNAs. For example: At the transcriptional level, lncRNAs can act as scaffold (guides) or as architects and enhancer-like lncRNAs, while at the post-transcriptional level, they play a very crucial role in mRNA processing. |
Response: Thank you for indicating important point. As reviewer mentioned, we described molecular functions of lncRNAs includes role of guide, scaffold, transcription, and post-transcription regulator (Lines 60-84).
Comment 4
|
It is not clear why the authors chose LOC441461 for further examination compared to the other genes in Table 1. |
Response: The reason why we choose LOC441461 was to investigate impact of lncRNAs in cancer. Although there were two lncRNAs were significant including LOC441461 and AK097446, the impact of LOC441461 in colon cancer was reported in contrast of AK097446. For these reasons, we decided to investigate impact and molecular function of LOC441461.
Comment 5
|
Which RNA-seq data units (RPKM?) were retrieved from xena browser? Please include it in the material and methods section as well as in Figure 1. |
Response: Thank you for pointing out the Figure 1 details. The unit of STAD RNA-seq data in xena borwser is log2(RPKM+1). As reviewer mentioned, we have added in Figure 1 and line 95 of the Materials and Methods section.
Comment 6
|
In figure 1 please change mRNA to RNA as LOC441461 is lncRNA. |
Response: Thank you for pointing our critical mistake. As reviewer commented, we have changed y-axis of Figrue 1A, and Figure 1B to “LOC441461 lncRNA expression (log2(RPKM+1))”.
Comment 7
|
Have you evaluated LOC441461 prognostic value concerning disease progression and overall survival in STAD TCGA dataset? It is highly suggested that the authors include a more extensive clinical analysis of LOC441461 using STAD TCGA dataset as well as other publicly available datasets (e.g GEO datasets) in order to strengthen its clinical significance. |
Response: Thank you for helpful advise about conducting survival analysis. Using TCGA databse, we conducted survival analysis with overall survival and progression free interval between the combination of T and M category which shows significant difference of expression in the results of logistic regression analysis (Table 2). The results showed significantly different prognosis between six groups, specifically, T1 and T4 with M0, M0 and M1 with T4 in overall survival. In progression free interval, T1 and T4 with M0 showed significant different prognosis. Survival analysis was visualized as Kaplan-Meior curve in Figure 1C and Figure 1D. These also have been described in results section, lines 286-293 and lines 297-301. Additional results of log-rank test have been added to Table S5.
Comment 8
|
In general, the Discussion section needs improvement, since the authors mainly discuss their results. Please include current publications and discuss more the translational impact of your study. |
Response: Thank you for reviewers’ suggestion, we agree that our discussion part is focused on our results. To avoid any confusion to the reader, we have removed those statement from the manuscript. We have instead stated with current publications and additional impacts on our study.
Comment 9
|
In the discussion section the authors should indicate the most significant limitations of their work. |
Response: Thank you for reviewers’ suggestion. We agree that there are limits to showing translational impact to validate. The requirement for further mechanistic study has been discussed in the Conclusion part of the revised manuscript (Line 552-556).

Round 2
Reviewer 1 Report
Only one thing goes wrong.
1. Line 121-123, "The cytoplasmic or nuclear location of LOC441461 was assessed in MKN74 cells using the PARIS kit (Thermo Fisher Scientific, Waltham, MA, USA) according to the manufacturer’s protocol.", Were only MKN74 cells used in this study? However, line 281-283, "(C) Subcellular fractionation was performed using the PARIS kit to separate the nuclear and cytoplasmic fractions of MKN74 and SNU216 cells.
Author Response
Response to Reviewer 1 Comments
ANSWERS TO THE REVIEWERS' COMMENTS
We thank you and the reviewers for the valuable comments on our manuscript. Our responses to the comments are provided below. The comments have substantially improved the quality of the manuscript. The revisions in the manuscript are marked up using the “Track Changes” function, and the relevant line numbers are provided with our response. The response of “Autor’s Reply to the Review Report (Reviewer 1)” has been written in red as an example.
Reviewer #1: Only one thing goes wrong.
Comment 1
|
Line 121-123, "The cytoplasmic or nuclear location of LOC441461 was assessed in MKN74 cells using the PARIS kit (Thermo Fisher Scientific, Waltham, MA, USA) according to the manufacturer’s protocol.", Were only MKN74 cells used in this study? However, line 281-283, "(C) Subcellular fractionation was performed using the PARIS kit to separate the nuclear and cytoplasmic fractions of MKN74 and SNU216 cells. |
Response: Thank you for pointing out our critical mistake. As shown in Figure 2C, subcelluar fractionation was performed using MKN74 and SNU216 cell lines. This has been corrected in lines 118-119.

Reviewer 3 Report
The authors have assessed the recommendations, however, there are some minor issues they have to address in order to be able to publish their results.
- Introduction needs editing. Syntax and grammar, especially in paragraph 2 (lncRNAs), should be improved.
- Please correct several typos throughout the manuscript. For instance, please correct "2.3 Survival analysis The sulvival analysis" to "2.3 Survival analysis The survival analysis"
- Figure 1 sharpness should be improved. Font size should be increased in Figure 1C & 1D. Please include in fig. 1 legend the statistical analysis performed for the survival analysis.
- The language of the manuscript should be edited by a native English speaker.
- The discussion needs improvement.
- In the conclusions of the present study, the statement "which has significant value in the diagnosis, development and prognosis of gastric cancer" should be revised.
Author Response
Response to Reviewer 3 Comments
ANSWERS TO THE REVIEWERS' COMMENTS
We thank you and the reviewers for the valuable comments on our manuscript. Our responses to the comments are provided below. The comments have substantially improved the quality of the manuscript. The revisions in the manuscript are marked up using the “Track Changes” function, and the relevant line numbers are provided with our response. The response of “Autor’s Reply to the Review Report (Reviewer 3)” has been written in red as an example.
Review Comments to the Author:
Reviewer #3: The authors have assessed the recommendations, however, there are some minor issues they have to address in order to be able to publish their results.
Comment 1
|
Introduction needs editing. Syntax and grammar, especially in paragraph 2 (lncRNAs), should be improved. |
Response: Thank you for helpful suggestion. We deeply agree that our revised manuscrpt should undergo extensive English revisions with reviewer’s comment. We have revised the manuscript using the editorial services listed at https://www.mdpi.com/authors/english with our editor's recommendation.
Comment 2
|
Please correct several typos throughout the manuscript. For instance, please correct "2.3 Survival analysis The sulvival analysis" to "2.3 Survival analysis The survival analysis" |
Response: Thank you for pointing out our critical mistake. As reviewer mentioned, the typo has been corrected as suggestion of reviewer in line 99.
Comment 3
|
Figure 1 sharpness should be improved. Font size should be increased in Figure 1C & 1D. Please include in fig. 1 legend the statistical analysis performed for the survival analysis. |
Response: Thank you for helpful advice. As reviewer mentioned, sharpness of Figure 1 has been improved. Specifically, the font size of Figure 1C and Figure 1D was increased than previous manuscript version. Additionally, the information of statistical analysis has been added in Figure 1C and 1D, and the Figure 1 legend has been added to lines 282-284.
Comment 4
|
The language of the manuscript should be edited by a native English speaker. |
Response: Thank you for helpful suggestion. We deeply agree that our revised manuscrpt should undergo extensive English revisions with reviewer’s comment. We have revised the manuscript using the editorial services listed at https://www.mdpi.com/authors/english with our editor's recommendation.
Comment 5
|
The discussion needs improvement. |
Response: Thank you for helpful suggestion. The discussion section has been improved with syntax and grammar during English editing service. In addition, the recent potential as an RNA biopharmaceutical has been added to lines 286-289.
Comment 6
|
In the conclusions of the present study, the statement "which has significant value in the diagnosis, development and prognosis of gastric cancer" should be revised. |
Response: Thank you for helpful advice. As reviewer mentioned, pointed statement has been revised in line 491-493.

Reviewer 4 Report
The authors have successfully addressed the suggested comments concerning their manuscript entitled “Downregulation of LOC441461 promotes cell growth and motility in human gastric cancer”. However, there are still some minor issues they should take into consideration in order to publish their results.
- English language is very poor. Extensive language editing by a native English speaker should be performed.
- There are many typos that the authors have to correct before publishing their manuscript.
- The authors have successfully evaluated LOC441461 prognostic value in STAD TCGA dataset. However, figure 1 quality is poor and not informative concerning Kaplan Meier curves.
Author Response
Response to Reviewer 4 Comments
ANSWERS TO THE REVIEWERS' COMMENTS
We thank you and the reviewers for the valuable comments on our manuscript. Our responses to the comments are provided below. The comments have substantially improved the quality of the manuscript. The revisions in the manuscript are marked up using the “Track Changes” function, and the relevant line numbers are provided with our response. The response of “Autor’s Reply to the Review Report (Reviewer 4)” has been written in red as an example.
Review Comments to the Author:
Reviewer #4: The authors have successfully addressed the suggested comments concerning their manuscript entitled “Downregulation of LOC441461 promotes cell growth and motility in human gastric cancer”. However, there are still some minor issues they should take into consideration in order to publish their results.
Comment 1
|
English language is very poor. Extensive language editing by a native English speaker should be performed. |
Response: Thank you for helpful suggestion. We deeply agree that our revised manuscrpt should undergo extensive English revisions with reviewer’s comment. We have revised the manuscript using the editorial services listed at https://www.mdpi.com/authors/english with our editor's recommendation.
Comment 2
|
There are many typos that the authors have to correct before publishing their manuscript. |
Response: Thank you for pointing out our mistakes. As reviewer mentioned, the typos have been corrected during English correction service.
Comment 3
|
The authors have successfully evaluated LOC441461 prognostic value in STAD TCGA dataset. However, figure 1 quality is poor and not informative concerning Kaplan Meier curves. |
Response: Thank you for helpful advice. As reviewer mentioned, sharpness of Figure 1 has been improved. Specifically, the font size of Figure 1C and Figure 1D was increased than previous manuscript version. Additionally, the information of statistical analysis has been added in Figure 1C and 1D, and the Figure 1 legend has been added to lines 282-284.
